A large cockroach from the mesosaur-bearing Konservat-Lagerstätte (Mangrullo Formation), Late Paleozoic of Uruguay

Calisto Viviana
Piñeiro Graciela fossil@fcien.edu.uy
Departamento de Paleontología, Facultad de Ciencias , Montevideo , Uruguay
Gillespie Joseph
Electronic publication date: 2019 Jan 18
Publication date: 2019
Volume: 7
Electronic Location ID: e6289
Received 2018 Aug 6; Accepted 2018 Dec 16
Copyright: ©2019 Calisto and Piñeiro
Copyright year: 2019
Copyright holder: Calisto et al.
License: This is an open access article distributed under the terms of the Creative Commons Attribution License, which permits unrestricted use, distribution, reproduction and adaptation in any medium and for any purpose provided that it is properly attributed. For attribution, the original author(s), title, publication source (PeerJ) and either DOI or URL of the article must be cited.
License URL: https://creativecommons.org/licenses/by/4.0/

Keywords: Insecta, Uruguay, Late Carboniferous-Early Permian, New species, Blattaria

Funding: ANII FCE_2011_6450 NGS grant 049714 PEDECIBA BIOLOGÍA to Viviana Calisto This work was supported by ANII FCE_2011_6450 and by NGS grant 049714 to Graciela Piñeiro, and by PEDECIBA BIOLOGÍA to Viviana Calisto. There was no additional external funding received for this particular study. The funders had no role in study design, data collection and analysis, decision to publish, or preparation of the manuscript.

==============================
Barona arcuata, n.gen et n.sp., a left forewing of a relatively large cockroach of the Order Blattaria, is described from mesosaur-bearing lagoonal shales of the Mangrullo Formation (north-eastern Uruguay). While most of the insect remains recovered from the Mangrullo Formation come from sandy limestones, associated to scarce isolated mesosaur bones and pygocephalomorph crustaceans, the cockroach wing here described was found in the overlaying green to brownish, gray and dark black shales associated to intercalated bentonites and evaporitic gypsum crystals. Barona arcuata shares some features with typical Late Carboniferous taxa such as its general venation pattern and outline of the wing, four main and powerful veins arising close together from near the base of the wing, Sc simple forked, pectinate, reaching the costal border through a long fork, R and M bifurcating and terminating in the wing margin above and below the apex respectively, short and narrow CuA, and the presence of a broad interspace between CuP and AA. Cross venation seems to be absent or it was not preserved. Some characters might relate Barona arcuata to the Late Carboniferous-Early Permian Neothroblattinidae such as the presence of sigmoidal veins in the anal area, a condition not found in any of the remaining representatives of the Palaeozoic Blattaria. Intriguingly, the Uruguayan blattarian also presents a strong similarity with Qilianiblatta namurensis Zhang, Schneider & Hong, 2012 from the Westphalian of China, clearly a smaller taxon that is also difficult to relate to any of the preexistent families. The apparent plesiomorphic venation pattern of the new species which is reminiscent of that present in the oldest known blattarians, is in agreement with a Permo-Carboniferous (Gzhelian-Asselian) age for the Mangrullo Formation also supported by the presence of a macrofloral assemblage dominated by arborescent lepidondendrids and other lycopsids and the pygocephalid-like morphology of the pygocephalomorph crustaceans from the same levels.

Introduction

In Uruguay, Late Paleozoic insects are overwhelmingly represented by isolated wings (Pinto, Piñeiro & Verde, 2000) found in calcareous levels of the Late Paleozoic Mangrullo Formation which crops at the northeast of the country, in the Cerro Largo County (Fig. 1). The insect wings come from levels containing pygocephalomoph crustaceans, silicified fragmentary trunks and scattered mesosaur remains (Piñeiro, 2002; Piñeiro, 2006; Piñeiro et al., 2012a; Piñeiro et al., 2012b). From these coarse to fine sandy limestone and succeeding grey-brownish shale facies, corresponding to a glacioeustatic sea falls (Mackinnon, De Santa Ana & Pessi, 1982), several well preserved imprints of isolated wings were described as belonging to Hemiptera belonging to Cicadopsyllidae Martynov, 1931, and to the new family Perlapsocidae Pinto et Piñeiro, 2000. Two new species were erected, Paracicadopsis mendezalzolai Pinto et Piñeiro, 2000 and Perlapsocus formosoi Pinto et Piñeiro, 2000, opening a new line of research for the Late Palaeozoic deposits of Uruguay.

Figure 1 Description of a new blattid from the Early Permian of Uruguay.

Geographic location of the insect bearing Mangrullo Formation. (A) Map of Uruguay showing the location of Cerro Largo county (in yellow) at north eastern Uruguay (modified from Piñeiro, 2004); (B) Photograph showing the black shale of the Mangrullo Formation at the El Baron locality. White arrow points the equivalent stratigraphic levels to those where the holotype of the Uruguayan cockroach was found; (C) Detailed map of the area of outcrops of the Mangrullo Formation. Pink asterisk points the location of the El Barón locality at the Cerro Largo County (modified from Piñeiro et al., 2012a). (B: photo credit: Graciela Piñeiro.)

At the time that Pinto, Piñeiro & Verde (2000) described those specimens representing the first Palaeozoic record of insects for Uruguay, the age of the Mangrullo Formation was controversial as most previous workers have placed these deposits into the Late Permian (Bossi & Navarro, 1991; Beri & Daners, 1995 among others). However, the hemipteran species described by Pinto, Piñeiro & Verde (2000) for the Mangrullo Formation were considered by these authors as comparable to components of the Early Permian Russian entomofauna (e.g., Becker-Migdisova, 1960). Several new insect remains showing an outstanding preservation of wings and some parts of the body were recently collected from the same sandy limestone levels where the cicadopsyllids appeared. Although these new specimens are currently under study (Calisto, 2018), it is possible to anticipate the presence of a moderately diverse insect fauna in the Mangrullo Formation limestone. However, the overlying grey to brownish metheorization shale containing articulated mesosaurs, very well preserved pygocephalomorphs (see Piñeiro et al., 2012a) and impressions of soft plant organs, have yielded the cockroach wing that we are describing herein (see Fig. 2), representing a promising level for collecting more insects and plants.

Figure 2 Description of a new blattid from Uruguay.

Sedimentological and stratigraphic framework of the El Baron type locality. (A) Grey-brownish shales of the Mangrullo Formation, intercalated by centimetric bentonitic levels. (B) Brouwnish silty shale facies of the Mangrullo Formation containing mineralized trunks and impressions/compressions of soft plant remains and insect wings. (C) Sandy dolomitic limestone and breccias of the Mangrullo Formation, representing shallow coastal environments during glacioeustatic regressions (see text for additional information). (D) Litho and biostratigraphic profile of the Mangrullo Formation at the El Baron locality where the new Uruguayan blattarian was found (represented by reddish-brown wings in the stratigraphic section). The white arrow indicates the levels that yielded the Barona arcuata forewing, while red arrows point to the levels where other insect groups have been discovered. (E) Legend. Modified from Piñeiro et al. (2012b). (2A-C photo credit: Graciela Piñeiro.)

Cockroaches are a phylogenetical, biostratigraphical and ecologically important order of insects (Vršanský et al., 2017). They originated and became dominant during the Carboniferous. They apparently give origin to Mantises in the Cretaceous and become a currently well represented group after 310 to 320 million years of evolution (Zhang, Schneider & Hong, 2012; Wei & Ren, 2013; Evangelista, Djernæs & Kaur Kohli, 2017; Vršanský et al., 2017). The earliest fossil record of cockroaches dates back to Late Carboniferous, and show evidence that tegminas have appeared early as an adaptation for protection (Zhang, Schneider & Hong, 2012). In the Paleozoic, nine extinct insect families have been recorded (Schneider, 1983; Vršanský et al., 2017); there are eight species of Phyloblattids in South America from the Itararé Group of Brazil (Carboniferous-Permian) (see Rösler, Rhon & Albamonte, 1981; Pinto, 1972a; Pinto, 1972b; Pinto & Pinto de Ornellas, 1978; Pinto & Pinto de Ornellas, 1980; Pinto, 1990; Ricetti et al., 2016) and from the Rio Genoa Formation (Early Permian) of Argentina (Ricetti et al., 2016). However, no blattids have been found in the Early Permian Iratí Formation (purposely correlative to the Mangrullo Formation, e.g., Bossi & Navarro, 1991) despite insects would have had a high preservational potential in this Konservat-lagerstätte (Silva et al., 2017). Here, we describe a left forewing preserved as part and counterpart that represents the first and only record of Blattaria for Uruguay. We also propose some hypotheses to explore the repercussion of this finding in the currently accepted biostratigraphical and biogeographical context of the Gondwanan region of Pangaea.

Materials and Methods

The single left forewing (part and counterpart) described herein (FC-DPI 8710) (Fig. 3) was found in shales at the El Baron locality (Mangrullo Formation, Cerro Largo County) and it is housed in the Collection of Fossil Invertebrates of the Department of Paleontology at Facultad de Ciencias-UdelaR, Montevideo, Uruguay (FC-DPI).

Figure 3 Description of a new blattid from the Early Permian of Uruguay.

FC-DPI 8710, Barona arcuata. Photographs of the left forewing, preserved as part (A) and counterpart (B). Scale bar: 10 mm; (C) Distribution and terminology of veins: C, costa; Sc, subcosta; RA, anterior radius; RP, posterior radius; MA, anterior media; MP, posterior media; CuA, anterior cubitus; CuP, posterior cubitus; AA, anterior anal; AP, posterior anal; CV, connecting vein.

The venation pattern of the holotype of Barona arcuata (FC-DPI 8710) was examined and drawn in dry state with the aid of a stereomicroscope with incorporated camera lucida (NIKON HFX-DX). Photographs were made directly using a NIKON digital camera under sided crossed light, and others were taken using the camera integrated to the stereomicroscope and processed with the software Infinity Analize for more detailed images. Some other photographs were taken using the light produced by a light table placed under the specimen. The venation pattern was determined from composite line drawings of part and counterpart, improved by using Adobe Illustrator CS6 software, and then the images were calibrated using the photograph scales. The wing venation pattern paradigm of Lameere (1923) was followed, along to references in Bethoux, Schneider & Klass (2011) for the radial vein system.

The new Uruguayan blattarian Barona arcuata was compared to other Gondwanan representatives of the group (Fig. 4) such as the Brazilian phyloblattid Anthracoblattina mendezi Pinto & Sedor (2000), from the Permo-Carboniferous Itararé Group (Ricetti et al., 2016), and also to several Carboniferous and Early Permian taxa from the Laurasian region of Pangaea, especially the Neorthroblattinidae and the Chinese taxon Qilianiblatta namurensis (Stephanian) (Zhang, Schneider & Hong, 2012; Guo et al., 2013), to which the Uruguayan Barona arcuata features the major anatomical similitude (particularly in vein distribution).

Figure 4 Description of a new blattid from the Late Palaeozoic of Uruguay.

Comparative venation distribution between the Uruguayan new blattid. (A) Barona arcuata, (B) the Chinese Qilianiblatta namurensis and (C) the Brazilian Anthracoblattina mendezi. B and C, adapted from Zhang, Schneider & Hong (2012) and Ricetti et al. (2016), respectively.

The electronic version of this article in Portable Document Format (PDF) will represent a published work according to the International Commission on Zoological Nomenclature (ICZN), and hence the new names contained in the electronic version are effectively published under that Code from the electronic edition alone. This published work and the nomenclatural acts it contains have been registered in ZooBank, the online registration system for the ICZN. The ZooBank LSIDs (Life Science Identifiers) can be resolved and the associated information viewed through any standard web browser by appending the LSID to the prefix http://zoobank.org/. The LSID for this publication is: urn:lsid:zoobank.org:pub:25614310-CE7B-4D77-9CEB-2CD5B93B7B2D

The online version of this work is archived and available from the following digital repositories: PeerJ, PubMed Central and CLOCKSS.

Barona: urn:lsid:zoobank.org:act:7CE17B81-0818-498C-87C1-937B75795F40

Barona arcuata: urn:lsid:zoobank.org:act:6A6269D2-A7EC-4EF2-B802-32AAFFBE69D4.

Geological Setting

The Mangrullo Formation crops up in north and northeast Uruguay (Fig. 1), extending to Brazil and thus forming part of the Paraná Basin as the correlative of the Iratí Formation (Bossi & Navarro, 1991). It was deposited in a restrict, variably hypersaline lagoon (Piñeiro et al., 2012b) formed under short-term regressive-transgressive glacioeustatic phases both probably linked to the Permo-Carboniferous Gondwanan glacial and interglacial cycles (Mackinnon, De Santa Ana & Pessi, 1982). Sandy dolomitic limestone and breccias bearing asymmetric ripple marks were deposited following the start of the regression event, and represent shallow, coastal environmental conditions and relatively more energetic episodes. They were favourable for the preservation of insect wings associated to pygocephalomorphs, mesosaur remains and silicified tree trunks (Pinto, Piñeiro & Verde, 2000; Piñeiro, 2006). Low energy and poorly oxygenate conditions favouring the development of highly fossiliferous black bituminous and grey to brownish shale-dominated facies, with intercalated bentonitic levels, which unconformably overlies the limestone (Fig. 2). These shales bear well-preserved leaf cuticles, stems and reproductive organs, insect wings, mesosaur skeletons and very nicely preserved pygocephalomorph crustaceans plus some perminerlized (probably silicified) tree-ferns. This assemblage represents a Konservat-Lagerstätte which is characterized by the exquisite preservation of the specimens, including very delicate soft tissues (Piñeiro et al., 2012a; Piñeiro et al., 2012b; Piñeiro et al., 2012c). A Late Permian age was initially proposed for the Mangrullo Formation based on palynological associations (Bossi & Navarro, 1991; Beri & Daners, 1995) following the same line of reasoning suggested for the Brazilian Iratí Formation (e.g., Daemon & Quadros, 1970; Mezzalira, 1980). Nonetheless, new biostratigraphic studies including macrofloral correlations and the presence of a mesosaurid-pygocephalomorph association present in both these units as well as in the South African Whitehill Formation, allowed the placement of all these strata into the Lower Permian (Huene, 1940; Oelofsen, 1981; Piñeiro, 2002; Piñeiro, 2006). More recent geochronological studies performed in zircons of the Iratí and Mangrullo bentonites obtained two different ages for these deposits, one of 278 ± 2 Ma (Artinskian) (Santos et al., 2006), 276 ± 5 and 270 ± 5 Ma (Rocha-Campos et al., 2006); 270 ± 5 to 279 ± 6 (De Santa Ana, Goso & Daners, 2006) and even more recently other data suggested an age close to the Permo-Carboniferous boundary (AC Rocha-Campos, pers. comm., 2014) confirming that even applying similar analytic methodologies, the resulting ages were not consistent.

Systematic Palaeontology

Class Insecta Linnaeus, 1758	
Superorder Dictyoptera Latreille, 1829	
Order Blattaria Latreille, 1810	
Family insertae sedis	

Barona Calisto and Piñeiro, new genus

Type species Barona arcuata Calisto and Piñeiro

Diagnosis. As for the type species (see below). The unique morphology of the anal field venation including the presence of sigmoidal veins, as well as the narrow and short CuA, the broad interspace between CuP and the first anal vein, and the scarce development of Sc, combined with the large wing size distinguishes Barona from other genera within Palaeozoic Blattaria.

Derivation of name. The generic name, treated as feminine, refers to the El Baron Ranch, where the type specimen was found.

Barona arcuata Calisto and Piñeiro, new species (Figs. 3–5)

Figure 5 New large cockroach from the Late Palaeozoic of Uruguay.

FC-DPI 8710, Barona arcuata left forewing. Photograph showing the distribution of the sigmoidal anal venation and the wide interspace between CuP and AA. Scale bar: 10 mm. (Photo credit: Graciela Piñeiro.)

Type material. FC-DPI 8710, Holotype deposited at the Facultad de Ciencias Fossil Invertebrates Collection (acronym FC-DPI) of Montevideo, Uruguay.

Type Locality and Age. El Baron Ranch, Cerro Largo County, from non-bituminous shale of the Mangrullo Formation Konservat-Lagerstätte, ?Late Carboniferous-?Early Permian of Uruguay (Figs. 1 and 2).

Charater preservation. The studied specimen is preserved in both part and counterpart, possibly favoured by pyrite precipitation, a common taphonomic feature found in the Mangrullo Formation (see Piñeiro et al., 2012b).

Diagnosis. An elongated left forewing (32 mm ×12. 5 mm) having C completely marginal at the anterior part of margin; elongate, coastal field wedge-shaped, which is wide at the base, but narrowing posteriorly towards a half of the wing length; Sc simply branched or pectinate, with four branches in 45° angle that reach the costal margin; R sigmoidal, forked into RA and RP, with five main bifurcated branches terminating at the wing apex; M forked into MA and MP with four main branches that bifurcate and reach the wing posterior margin close to the apex; RA and MA differentiated; Cu divided into CuA and CuP, near the wing base; CuA simply curved, short and narrow, bifurcated into only four or five terminal branches that are not extended posteriorly; CuP very well-marked and sharply arcuated; connecting vein CV arculus (sensu Bethoux, 2005) or archedyction absent or not visible; R, M, and CuA strongly developed; AA and AP with few branches including sigmoidal venation in the middle area; wide interspace between CuP and AA venation present.

Derivation of name. The species name is derived from the Latin arcuata (in feminine gender) and refers to the arcuate feature of the CuP vein.

Description

Left forewing elongate, ellipsoid, length two times and a half longer than its width (32 mm × 12. 5 mm) (Fig. 3). The wing bears a marginal C and an elongated, wedge-shaped coastal area, which is broad near the base and narrows posteriorly until the middle of the wing length. The four main veins (Sc, R, M, Cu) arise from the wide, three dimensional, and strongly sclerotized wing base and bifurcate posteriorly into multiple branches. The Sc is simply bifurcated (pectinated) with at least four branches coming off from the main stem at 45° angles, the last two longer and reaching the wing margin; R sigmoidal, clearly divided into extensive RA and RP with five main branches bifurcated into multiple veins terminating at the wing margin, just above the wing apex. First branch bifurcates once, the following two bifurcate twice and the last three bifurcate three times. M forked into MA and MP, with 19 bifurcated branches extending to wing margin just below the apex; RA and MA are weakly differentiated; Cu curved posteriorly and divided into CuA and CuP; CuA almost sigmoidal and very narrow with four or five branches; CuP vein is very well-marked and follows a sharply arcuate direction to the internal border of the tegmina; connecting vein CV arculus (sensu (Bethoux, 2005) between M and CuA is absent or not preserved; AA and AP with numerous branches that do not reach the ventral border, cross venation might be present in middle and basal area of wing, but the preservation of the only available specimen does not allow us to be sure about the state of this character (Fig. 5).

Comparisons

Phyloblattidae—The venation present in Barona arcuata seems to share some characters with the family Phyloblattidae (sensu Schneider, 1983): costal area elongated and comparatively narrow, though it is basally wider (wedge-shaped) in the Uruguayan taxon, Sc pectinate, but with simple, not forked few branches in Barona, RA and MA weakly differentiated, but with MA, instead RA more developed in Barona, sigmoidal CuA, but constrained to a small area near the CuP vein in Barona, while well extended to the apex in Phyloblattids), and the presence of a wide space between CuP and AA veins (sensu Schneider, 1983; Schneider, Lucas & Scholze, 2017; J Schneider, pers. comm., 2018), which appears to be notably narrower in Phyloblattids (Fig. 4). Moreover, Barona lacks transversal venation, and a scalariform cross venation and some reticulate areas are not well visible from the preserved currently unique specimen (Fig. 5).

Mylacridae: Neorthroblattina—Barona also shares some characters with Mylacridae, particularly with Neorthroblattina (Schneider, 1978; Schneider, 1980; Schneider, Lucas & Scholze, 2017) such as the wedge-shaped costal area, the pectinate Sc bearing just a few veins, the CuA with few branches constrained to a small field near the CuP vein (although there is a tendency of the vein forking growing posteriorly in neorthroblattinids), and the sigmoidal arrangement of the anal veins. However, Barona seems to lack the extensive cross venation observed in neorthroblatinids and other mylacrids, and it is frankly larger.

Poroblattinidae—Comparing with Poroblattinidae, Barona shares the sigmoid form of R and M, a short CuA vein, and the general arrangement of anal veins, but differs from representatives of this family because they present a short Sc, the space between CuP and AA veins is narrow, the wing size does not exceed 10 mm and the shape of the wing is rounded (Schneider, 1978).

Barona shares with Poroblattinids and Mylacrids the wedge-shaped morphology of the costal area, but the CuA in these taxa is straight while in Barona and Archimylacris (Ross, 2010) it is smoothly curved.

Incertae sedis blattarian from China—Intriguingly, the Uruguayan cockroach is very similar to Qilianiblatta namurensis (Zhang, Schneider & Hong, 2012) from the Stephanian of China, sharing the following characters that differentiate it from the Phyloblattids here represented by Anthracoblattina (see Fig. 4):

(a) Costal area is narrow but basally wider, while it is uniformly narrow in Anthracoblattina.

(b) RA and RP are weakly differentiated, although yet conserving the plesiomorphic arrangement, lacking translocations (Guo et al., 2013). These veins appear as no clearly differentiated in Anthracoblattina.

(c) Anal field is well delimitated by first AA, which is well separated to the deeply incised and arquate CuP in Barona and Qilianiblatta; this last vein is discreet in Anthracoblattina and the space between CuP and AA seems not as broad as in Barona and Qilianiblatta. The anal field in Barona resembles more the pattern in Neorthroblattina (Schneider, 1978; Schneider, 1980), but it is not as well preserved as the other main venations.

(d) Barona and Qilianiblatta additionally differ from Anthracoblattina in their CuA, which seems to be slightly curved and smoothly sigmoid, occupying a small area near CuP, whereas it is almost straight in Anthracoblattina and extends further to almost reach the wing apex.

Although Barona, Qilianiblatta and Anthracoblattina all bear relatively large forewings, the latter is substantially larger (42 mm against 32 of Barona and 18 to 25 of Qilianiblatta). All of them present a very different morphology from the main pattern seen in Permian blattids (see Vršanský & Aristov, 2012).

Discussion

Cockroaches are part of the Mangrullo Formation entomofauna, confirming the wide distribution of this group in South America. The Uruguayan Mangrullo Formation represents an ancient Konservat-Lagerstätte preserved in a moderately hypersaline lagoon, under poorly oxygenated bottom conditions, an environment with a high potential of soft tissues preservation (Piñeiro et al., 2012b). Most of the insect wings collected until now (Pinto, Piñeiro & Verde, 2000; Calisto, NuñezDemarco & Piñeiro, 2016) are referred to orders known to have aquatic or semiaquatic adaptations, such as hemipterans and coleopterans (Wallace & Anderson, 1995). There is evidence that in the Permian, some schizophoroid beetles went adapted to the aquatic habitat (Sinitshenkova, 2002), and Mesozoic beetles lived on the surface of the water but did not swim (Ponomarenko, 2003). Barona arcuata and other families recognized to be present in the Mangrullo Formation (Calisto, 2018) surely had terrestrial habits considering that the oldest aquatic cockroach, even winged forms, are known from the Jurassic (P Vršanský, pers. comm., 2018).

Barona arcuata constitutes the first and oldest record of Blattaria for Uruguay, it shows several plesiomorphic characters that are typical of Carboniferous cockroaches (vide (Sellards, 1906), such as four main veins arising from the anterior end of the wing (Fig. 3); the front veins, particularly the subcostal extending beyond the middle length of the wing, where meets the margin throw a long fork; the radial and the median veins extend to the apex and cover the anterior and posterior area respectively, the cubital veins are simple and reach the posterior (medial) margin close to the middle length of the wing. All the cubital veins are arcuate and the CuA vein is bifurcated once before to reach the posterior (medial) margin; a well-defined anal area, with thick and arched anal margin. Moreover, the subcostal vein is basally separated from the radial vein, a characteristic observed in the oldest known, Carboniferous blattarians. According to this particular morphologic arrangement or combination of features, the Uruguayan taxon is considered to belonging to the new genus and species, Barona arcuata.

Permian cockroaches are distinguished by the fusion of R and M, in a way that only three principal veins reach together from the anterior end of the wing (Sellards, 1904; Sellards, 1906). That fusion is not observed in Barona arcuata, which thus appears to be similar in that condition to other ancient South American taxa as for instance Anthracoblattina, specially to A. mendesi (Ricetti et al., 2016) from the purposed earliest Permian (or Permo-Carboniferous) Campáleo area, State of Santa Catarina, Brazil, and also to Anthracoblattina archangelsky from the Carboniferous Rio Guenoa Formation of the Chubut Province, Argentina (Pinto & Mendes, 2002). They also share the large anal field, and its large size (Fig. 4). However, the morphology of the costal field and the distribution of some of the main venations are different; while in Barona the costal field broadens anteriorly, it becomes narrower in Anthracoblattina. Similarly, those branches of the Anal area in Barona are not regularly spaced, not all appearing to reach the posterior (medial) margin and the presence of sigmoidal anal venation characterize this taxon; CuA is slightly curved to the inner margin, instead straight; the number of MA and MP branches is not equivalent; the first fork for R is placed near the basal end and the presence of the connecting vein CV, which is absent in the Anthracoblattina species (see Fig. 4). Comparisons to the other Carboniferous and Permo-Carboniferous taxa from Brazil are difficult because of the fragmentary nature of the specimens, and much of them would require a detailed previous revision and redescription (cf. Ricetti et al., 2016).

On the other hand, the Uruguayan cockroach cannot be related to any of the described families within Blattaria and for the erection of a new family we would have to find additional specimens that allowed a correct calibration of the diagnostic characters. Even though it is intriguingly similar to Qilianiblatta namurensis from the Namurian-Westphalian (Carboniferous) of China, which is the oldest roach recorded at this moment that was also not assigned to a previously known family within order (Zhang, Schneider & Hong, 2012; Guo et al., 2013). The diagnosis of the Uruguayan new taxa includes many coincident conditions with the Chinese blattarian, as for instance the pattern of C, Sc, RA, RP, MA, MP, CuA and CuP venation. Even so, they varied in the number of vein branches and the absence of cross venation in the Uruguayan specimen. The latter may have been originally more widely distributed in Barona but it is masked by weathering or was destroyed during fossilization. Like the Chinese blattarians, Barona arcuata has the CuP arcuate and the base of the wing much sclerotized to displays primary dichotomy of veins, which is less developed; these characters positioned Barona close to the primitive taxa and the oldest known cockroaches. The main differences are the size (length 32 mm in Barona arcuata against 25 mm as maximum known in Qilianiblatta), the distribution of anal venation and the number of branches present in the main veins, a character that could be intraspecifically variable (Bethoux, Schneider & Klass, 2011). Size variation can be within the range of one species, or may be related to sexual differentiation, as may be in the case of Qilianiblatta namurensis, where apart from the holotypic specimen, other smaller and even more complete individuals are known (see Guo et al., 2013). Although intraspecific body size variation in cockroaches can be very high (Cornwell, 1968; Roth, 1990), we consider that the Uruguayan blattarian is larger than the Chinese Qilianiblatta. Moreover, we consider that the geographic and possible stratigraphic distance between the Uruguayan cockroach and those species from China, along with its unique character combination (Fig. 3), support the erection of a new taxon for the Mangrullo Formation.

The groundplan observed in Barona includes a combination of plesiomorphic and apparently more derived characters within Blattaria (Schneider, 1983) (as the presence of sigmoidal anal veins and the putative lack of cross venation), a condition that was also recognized for the Chinese species, the later having being considered as possessing the plan from which all the known phyloblattids (and even necymylacrids; P Vršanský, pers. comm., 2018) could have evolved (Zhang, Schneider & Hong, 2012).

The age of the Mangrullo Formation revised in the light of the new fossils. –The age of the Mangrullo Formation has remained uncertain for several decades, when geologists and even palaeontologists considered that it was deposited in the Middle or even at the end of the Permian according to palynozonation (Beri & Daners, 1995). Later, when other fossils started to appear, they revealed an older age to these strata (Piñeiro, 2002), and new studies involving SHRIMP U-Pb zircon dating from the bentonite layers (Santos et al., 2006) and a review of the palynobiostratigraphy in the Paraná Basin (see Souza & Marques-Toigo, 2003; Souza & Marques-Toigo, 2005) revealed a more precise age for the Iratí and its correlative Mangrullo Formation within the Early Permian (Artinskian). More recently, particular taxa of the ancient Konservat-Lagerstatte, described for the Mangrullo Formation show more similarities to conspicuous components of Late Carboniferous, rather than to Early Permian assemblages elsewhere. For instance, the pygocephalomorph crustaceans are related to families mainly represented in sequences of Late Carboniferous age from North America (Brooks, 1962) and Europe (Schram, 1979), while rare Pygocephalidae findings have been also described for the Petrolia Formation thought to be Leonardian in age (Hotton et al., 2002), Pygocephalidae and Tealliocarididae are essentially Late Carboniferous families and the Gondwanan pygocephalomorphs seems to be more related to these families (Taylor, Shen Yan Bin & Schram, 1998; Piñeiro et al., 2012a).

Plant associations can be good evidence for chronostratigraphic correlation among strata, particularly when comparing large groups. Provincialisms do exist but in several regions we can found closely related taxa in very distant regions, such as the presence of typical Euramerican-like forms in Gondwana floras as such is the case of Stigmaria-like rhizomes and tree ferns (Iannuzzi & Pfefferkorn, 2002).

Western Gondwana experienced a clockwise movement from high to low latitudes during the Carboniferous (Eldridge, Scotese & Walsh, 2000); vide (Iannuzzi & Pfefferkorn, 2002). Low latitude geological sequences are favourable to plant preservation, even more under reducing conditions (Wagner, 2003). Such is the case of the Mangrullo Formation where impressions of compressed cuticles, organs and permineralized trunks are commonly found in the mesosaur-bearing shale levels (Fig. 2). These plants show affinities to species that are components of the Phyllotheca- Gangamopteris flora (Piñeiro, 2006; Christiano-de Souza et al., 2014); Calamites, Paracalamites, Schizoneura, Annularia, Cordaites, Sphenophyllum leafs, along to occasional permineralized trunks of Equisetales and Lepidondendrales (e.g., Stigmaria, Lepidodendron, and Walkia) (J Broutin, pers. comm., 2018) are preliminarily recognized. These plants are found associated to insects, partial mesosaur skeletons and almost complete pygocephalomorph remains. This ‘Carboniferous-like’ scenario could be explained by the conservative behaviour of the Gondwana floras since the late Carboniferous to the Early Permian warm-temperated interval, but there are new lines of evidence that may suggest other hypotheses. Three geographically well delimited floral provinces can be recognized in the Carboniferous and the Early Permian, the tropical Euramerican, the temperated Angara (North Africa) and the also temperated Gondwanan provinces (DiMichele, Pfefferkorn & Gastaldo, 2001). Within each province, plants and insects should be adapted to the prevalent environmental conditions, particularly influenced by the changing climate. There were similar climatic and environmental conditions in the Euramerican provinces than in Gondwana during the Carboniferous and also, the Late Carboniferous climatic conditions prevailed into the Early Permian (DiMichele, Pfefferkorn & Gastaldo, 2001). Thus, this can explain the presence of taxonomically equivalent floral assemblages throughout Euramerican and Gondwanan floras (see below).

Until recently, two typical floras were recognized in Gondwana; the Early Carboniferous lycopsid-dominated and the poorly diverse Late Carboniferous flora characterized by the presence of Nothorhacopteris and Botrychiopsis, but there are now new Carboniferous floras described that represent a combination including taxa from other regions of Pangaea (Iannuzzi & Pfefferkorn, 2002).

Other studies have focused on the collapse of the Euramerican tropical forests that is verified at the Desmoinesian–Missourian boundary (early Kasimovian, ∼307 Ma) which is evidenced by the disappearance of the Lycospora-producing lepidodendrids, and some other lycopsids (Falcon-Lang et al., 2018). One of the probable causes affecting these lycopsids is a short-term episode of aridification near the Desmoinesian–Missourian boundary that stressed the hydrophilic lepidodendrales (Falcon-Lang et al., 2018).

Lycopsids are present in the floral assemblage of the Mangrullo Formation, and sedimentological and geochemical, as well as palynological data suggest temperate, but moderately seasonal and xeric conditions under a scenario of periodic volcanic eruptions that affected a large area, including the Uruguayan Norte Basin and almost all the Paraná Basin (Santos et al., 2006). Layers of crystalline gypsum intercalated in the shale facies overlying the limestones proved the deterioration of environmental conditions to the implantation of arid and possibly colder climates (Falcon-Lang et al., 2018) that affected the flora and also the fauna at the Mangrullo Lagoon (Piñeiro et al., 2012b). The presence of Lycopsids, a Late Carboniferous key taxon (according to Falcon-Lang et al., 2018) in the Mangrullo Konservat-Lagerstätte can suggest an older age to these strata or its survivorship into the Latest Carboniferous or the earliest Permian.

Many no yet described insect wings from the Mangrullo Formation advisor the presence of a moderately diverse entomofauna, as is typical of temperate climate that may include basal representatives of several successful insect families (Calisto, 2018).

Therefore, was the Mangrullo Formation a refuge where Carboniferous communities survived into the Permian? Or it represents an older assemblage than previously thought? Perhaps new fossil discoveries and geochronological studies involving zircon dating from the several bentonitic levels intercalated within the shale will allow for a better constraint of the age of the Mangrullo Formation and its Konservat-Lagerstatte.

Paleobiogeographic considerations. –It is interesting to remark that the most common Late Paleozoic insects around the world are cockroaches. During the Triassic, they were as common to represent 30% of the total fossil insect record, decaying their diversity at the end of the Mesozoic Era (P Vršanský, personal comment). It means that most primitive cockroaches possessed a high dispersion rate and also a high resistance to transport, adapting the forewings as protective “elytra” (Schneider, Lucas & Scholze, 2017). According to Schneider, Lucas & Rowland (2004) cockroaches are proven to be good stratigraphic and paleoecologic tools; they were adapted to several environments, including marginal lagoonal settings (Schneider & Werneburg, 1993). In particular, ancient groups are represented mostly in the Carboniferous and also in the Early Permian of Euramerian (Schneider, 1983; Broutin et al., 1990; Schneider & Werneburg, 1993; Hmich et al., 2003; Hmich et al., 2005) as well as in the Carboniferous and Permo-Carboniferous of the Gondwanan South American entomofaunas (see Pinto, 1972a; Pinto, 1972b; Pinto & Pinto de Ornellas, 1978; Pinto & Pinto de Ornellas, 1980; Pinto, 1990; Pinto, Maheshwari & Srivasta, 1992; Pinto & Mendes, 2002; Ricetti et al., 2016). Recent discoveries of blattarians in the Souss Basin of northern Africa (Morocco sequences) suggested a comparatively older age for these deposits, within the Westphalian (Hmich et al., 2003; Hmich et al., 2005). Thus, there are closely related blattarians also in the Permo-Carboniferous of Euramerian as well as in Gondwana, which are associated to a mixed flora containing typical Carboniferous taxa associated to species that are more frequent in the Permian entomofaunas. The Westphalian plants and insects found in Morocco are closely related to those present in Carboniferous series of Europe, while the Early Carboniferous flora (Visean-Namurian) of western Africa (Niger) contains only Gondwanan representatives. Thus, according to Broutin et al. (1990); Broutin et al. (1995) it should be during the Middle Permian (Kungurian to Wordian) that these last floras were mixed by inclusion of earliest Permian (or perhaps Late Carboniferous) Euramerican taxa. Therefore, Broutin et al. (1990) and Broutin et al. (1995) suggested that there could have been a first invasion of Euramerican floral elements into Gondwana during the Early Permian, although the presence of some taxa that were extinct in the Late Carboniferous may suggest that there could have been more than one dispersive event. It is clear also that some migration of Gondwanan representatives to Euramerian has occurred via Morocco, as there is evidence of mixed floras in the Permian of southern Spain (Hmich et al., 2003). Consequently with the flora migration, similar insect dispersion is expected, given the long intimate interaction shown by these groups since their earliest evolution. Even though the original dispersal center is not easy to determine, it is possible that these evolutionary biogeographic patterns were constrained by the gradual but influent formation of Pangaea. The presence of cockroaches and other insect groups in the Late Carboniferous or Permo-Carboniferous strata of Brazil and possibly in the Latest Carboniferous-Earliest Permian of Uruguay may support the hypothesis that the long-term and particularly intense Gondwanan glaciations occurred in the Early (Late Visian) rather than the Late Carboniferous, as has been demonstrated by geochemical and paleomagnetic previous studies (see Caputo et al., 2008; Barham et al., 2012). Nevertheless, at least two other short-term, glacio-eustatically controlled regressions occurred near the end of the period producing successive temperate and humid conditions followed by evaporitic arid and colder climates (Falcon-Lang et al., 2018). This is also congruent with the seasonally and arid climate suggested by the macro and microfloral components of the Mangrullo Formation (Piñeiro, 2006) and with the paleogeographic scenario of a completely building Pangaea during the earliest Permian or Permo-Carboniferous times.

Conclusions

The new species Barona arcuata from the Mangrullo Konservat-Lagerstätte represents the first record of Blattaria for Uruguay and one of the oldest records of Gondwana. Like Carboniferous blattarians from China, the new Uruguayan taxon has a conservative venational groundplan respect to Permian blattids, although their familiar affinities will remain in discussion until more specimens of this taxon can be found. The particular combination of characters present in Barona, is not observed in any of the pre-established families. It seems to share most features with basal neorthroblattinids and archymylacrids, but apparently lacks a characteristic scalariform cross venation present in the Westphalian representatives. The closely similar morphology of the Uruguayan specimen to Qilianiblatta namurensis from the Westphalian of China is intriguing, but taking into account the biostratigraphic and paleobiogeographic aspects here discussed, along with the taxonomic structure of the associated floral and faunal components of the Mangrullo Lagerstätte, the presence of Barona might support an older age for these strata, close to the Carboniferous-Permian boundary. Otherwise, the Late Carboniferous fossil record of the Chinese-like blattarians (maybe components of a new family) would extend its geographic and stratigraphic record into the lowermost Permian.

We thank the owners of the El Baron Ranch, Mónica, Alec and Richard Hastings for their constant support of our research team. Richard Hastings revised the language for improving the grammar. We are grateful to Peter Vršanský and Dominic Evangelista for their useful and constructive reviews and we wish to thank also very much the kindly help received from Joerg Schneider, providing us of important bibliographic material and critical commentaries that highly improved our manuscript. GP wants to specially thank Prof. Irajá D. Pinto, even now in memory, for his kind friendship and the patience and dedication to spread all his knowledge to who initiated the long way to be a scientist. From what I learned from Irajá, Uruguay has currently an important collection of Carboniferous-Permian insects and pygocephalomorph crustaceans. Thank you, dear Professor.

Additional Information and Declarations

Competing Interests

Author Contributions

Data Availability

New Species Registration

Graciela Piñeiro is an Academic Editor for PeerJ.

Viviana Calisto conceived and designed the experiments, performed the experiments, analyzed the data, contributed reagents/materials/analysis tools, prepared figures and/or tables, authored or reviewed drafts of the paper, approved the final draft, this article is part of the MSc. thesis of Viviana Calisto.

Graciela Piñeiro conceived and designed the experiments, analyzed the data, contributed reagents/materials/analysis tools, prepared figures and/or tables, authored or reviewed drafts of the paper, approved the final draft.

The following information was supplied regarding data availability:

We have no Raw data for this article.

Specimens are housed Collection of Fossil Invertebrates of the Department of Paleontology at Facultad de Ciencias-UdelaR, Montevideo, Uruguay (FC-DPI). Specimen numbers are: FC-DPI 8710; FC-DPI 8710.

The following information was supplied regarding the registration of a newly described species:

Publication LSID: urn:lsid:zoobank.org:pub:25614310-CE7B-4D77-9CEB-2CD5B93B7B2D

Genus name: Barona

urn:lsid:zoobank.org:act:7CE17B81-0818-498C-87C1-937B75795F40

Species name: arcuata

urn:lsid:zoobank.org:act:6A6269D2-A7EC-4EF2-B802-32AAFFBE69D4

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
