# Peer review of "A large cockroach from the mesosaur-bearing Konservat-Lagerstätte (Mangrullo Formation), Late Paleozoic of Uruguay"

_PeerJ, doi:10.7717/peerj.6289_

## Round 0.1 · original submission · Major Revisions

Dear Drs. Calisto and Piñeiro:

Thanks for submitting your manuscript to PeerJ. I have now received two independent reviews of your work, and as you will see, the reviewers raised some concerns about the research. Despite this, these reviewers are optimistic about your work and the potential impact it will have on the entomological community. Thus, I encourage you to revise your manuscript accordingly, taking into account all of the concerns raised by both reviewers.

Importantly, please note that both reviewers have provided a marked-up copy of your manuscript.

In your revision, please ensure that an English expert has thoroughly assessed you writing and grammar. Also, both reviewers have indicated that important references are missing, so please address this. There also seems to be some missing taxonomic information and an incomplete assessment of the classification/terminology. Please define all necessary terms and taxa, and remain consistent with usage throughout your work. This includes assessing and describing fossil work and comparing this to modern specimens/taxa.

I look forward to seeing your revision, and thanks again for submitting your work to PeerJ.

Good luck with your revision,

-joe

·

Basic reporting

- Language seems to me good, although I am not native
- There is a substantial lack of modern literature (all is recommended in the file) - to demonstrate number of families is dated at 2013, while dozens of works have been published since that time (all are recommended in the file); also the most important papers on the fossil cockroaches are not cited (all are recommeded in the attached file); not cited is also the paper by Schneider who revise the group Neorthroblattinidae, which I am sure the taxon belongs to
- Structure is OK
- OK

Experimental design

no comment

Validity of the findings

- There are some minor concerns such as deformity presence, lack of genus description and others in the attached file. But article must be written according to the recommended alternative family determination.
--- Other OK

The article is precise and really important. I am sure authors can study the recommended literature, especially Schneider 1980 and compare all known (there are in this revision) Neorthroblattinidae. Also I am sure that authors will modify conclusion (fortunately it is also a primitive group so most, but not all conclusions are valid) and prepare a paper that will be a significant contribution to the field and out of it. Nevertheless, after very negative experience of revisions in this journal (authors did not respected revision in spite of my deep trust) I insist on seeing the manuscript again before final acceptance.

Additional comments

The article is precise and really important. I am sure authors can study the recommended literature, especially Schneider 1980 and compare all known (there are in this revision) Neorthroblattinidae. Also I am sure that authors will modify conclusion (fortunately it is also a primitive group so most, but not all conclusions are valid) and prepare a paper that will be a significant contribution to the field and out of it. Nevertheless, after very negative experience of revisions in this journal (authors did not respected revision in spite of my deep trust) I insist on seeing the manuscript again before final acceptance.

In the attached file, each color field contains comments (as previously not all authors saw that comments)

·

Basic reporting

Much of the work is written well and information clearly stated. However, there are sections where this is not true and the paper should be revised by a native english speaker. I have provided some feedback in the attached document.

The authors have done a good job of ascertaining a reasonable phylogenetic position for the discussed fossil and thus their usage of citations is fairly thorough, with a few select cases where statements need to be cited.

The structure of the article is fine overall but needs some reorganizing. Some small sections should be swapped between the methods and results. Within the taxonomy section the diagnosis, comparisons and description sections are a bit confused and need to be largely rearranged.

Overall, the paper discusses relevant issues, although I disagree with the discussion of the age of the geological context based on this fossil.

Experimental design

The research is within the scope and aims of the journal.

The research question is well defined, relevant to the data collected, and meaningful.

The investigation seems to clearly be performed with a high ethical standard.

The methods are described well.

The technical standard by which the investigation is carried out could be improved upon. There are two main issues.

1. While the investigators have done a good job of giving some systematic context for the fossil described this needs to be improved upon. Primarily, usage of the terminology "Blattodea", "cockroach", "blattid" and lack of the usage of the terms "Stem-Dictyoptera", and "roachoid" make the paper confusing at best and deeply misleading at worst. There is a debate about the positions of the families Phyloblattidae, Archimylacridae and other Permian roachoids. This, combined with vastly different definitions of the term "Blattodea" mean that the terminology used in this paper, without clarification, could be deeply misleading to the reader. I have given extensive references and explanation in the attached text.

I will comment colloquially on the usage of Blattodea, cockroach, blattid and other names here. All three of these terms refer to different things and none of them refer to the organisms you are describing here in this paper.

crown-Blattodea - Is an order of insects containing the extant superfamilies: Blaberoidea, Corydioidea and Blattoidea. Blattoidea contains numerous families, including Blattidae, as well as the Epifamily Termitoidae (the termites). So, Blattodea contains cockroaches AND termites.

Stem-Blattodea - contain all crown-Blattodea as well as other extinct cockroaches forming ootheca and not bearing an external ovipositor.

Cockroaches - this term refers to a paraphyletic assemblage of insects within stem and crown Blattodea that retain most of the following plesiomorphic conditions of: dorso-ventrally flattened body, lateral expanded pronotum, cursorial legs.

Blattid - this is a shortening of Blattidae, a common, extant family of cockroaches.

Mantodea - is the clade of insects that are sister to Blattodea

Crown-Dictyoptera - is a clade of insects containing both Mantodea and Blattodea.

Stem-Dictyoptera - is a clade of insects containing Mantodea, Blattodea and roachoid fossil taxa bearing an external ovipostor and other plesiomorphic characters.

Roachoid - a stem-dictyoperan; an insect appearing to have the body form of a modern cockroach (dorso-laterally flattened body, cursorial legs, wide pronotum) but bearing conditions plesiomorphic to crown-Dictyoptera (e.g. external ovipositor)

These distinctions are extremely important considering the debate about the position of such taxa. See Hornig et al and Evangelista et al for a short summary of the discrepancy.

Hörnig, M.K., Haug, C., Schneider, J.W. & Haug, J.T. 2018 Evolution of reproductive strategies in Dictyopteran insects — clues from ovipositor morphology of extinct roachoids. Acta Palaeontol. Pol. 63, 1-24.

Evangelista, D.A., Djernæs, M. & Kohli, M.K. 2017 Fossil calibrations for the cockroach phylogeny (Insecta, Dictyoptera, Blattodea), comments on the use of wings for their identification, and a redescription of the oldest Blaberidae. Palaeontol. Electronica 20.3, 1-23.

Below is a list of citations following this terminology. In the next comment I will explain further:

Wang, Z., Shi, Y., Qiu, Z., Che, Y. & Lo, N. 2017 Reconstructing the phylogeny of Blattodea: Robust support for interfamilial relationships and major clades. Sci. Rep. 7, 1-8. (doi:10.1038/s41598-017-04243-1).
Legendre, F., Nel, A., Svenson, G.J., Robillard, T., Pellens, R. & Grandcolas, P. 2015 Phylogeny of Dictyoptera: Dating the origin of cockroaches, praying mantises and termites with molecular data and controlled fossil evidence. PLoS One 10, e0130127. (doi:10.1371/journal.pone.0130127).
Djernæs, M., Klass, K.D. & Eggleton, P. 2015 Identifying possible sister groups of Cryptocercidae+Isoptera: A combined molecular and morphological phylogeny of Dictyoptera. Mol. Phylogenet. Evol. 84, 284-303. (doi:10.1016/j.ympev.2014.08.019).
Ware, J.L., Litman, J., Klass, K.-D. & Spearman, L.A. 2008 Relationships among the major lineages of Dictyoptera: The effect of outgroup selection on Dictyopteran tree topology. Syst. Entomol. 33, 429-450. (doi:10.1111/j.1365-3113.2008.00424.x).
Inward, D., Beccaloni, G. & Eggleton, P. 2007 Death of an order: A comprehensive molecular phylogenetic study confirms that termites are eusocial cockroaches. Biol. Lett. 3, 331-335. (doi:10.1098/rsbl.2007.0102).
Beccaloni, G. 2018 Cockroach Species File Online. Version 5.0/5.0. (World Wide Web electronic publication, World Wide Web electronic publication.
Beccaloni, G. & Eggleton, P. 2013 Order: Blattodea. Zootaxa 3703, 46. (doi:10.11646/zootaxa.3703.1.10).

The discrepancy here comes from the discordance between fossil researchers and researchers on extant taxa. Grimaldi & Engel 2005 mention that Blattodea is a word used for Palaeozoic “roachoid” lineages and Blattaria is the ordinal term for extant cockroaches (at the time termites were not necessarily considered to be part of Blattaria). However, Inward et al. 2006 uses Blattodea for extant cockroaches, which follows Brunner von Wattenwyl. 1882 who was the first to say that extant cockroaches were an “order” rather than a "family" of Orthoptera. Since Inward et al. 2006 was an important paper, most people since have followed and use Blattodea in the sense of Brunner von Wattenwyl. 1882 and not that discussed by Grimaldi and Engel 2005. However, many fossil cockroach workers (but not all) ignore this and use other terminologies.

My opinion is that calling roachoids “Blattodea” is not in line with the current state of the art and widens the divide between palaeoentomologists and extant-entomologists. It is confusing to those who are unaware of the historical back and forth between the usage of these names, which is why I definitely think any palaeozoic roachoids should not referred to as Blattodea. I also think they should not be called cockroaches, but you could argue that they should be. If you want to call them cockroaches i would ask that you justify it in the paper using morphological characters.


2. A more recent set of tegmina vein homologies should be used. Although, to my understanding, Lameer is an important source that is still followed today (to some degree), there have been updates to this system, in particular by Guo et al 2013. You should follow them or a number of other Dictyoptera specific publications that would be relevant:
Guo, Y., Béthoux, O., Gu, J.-j. & Ren, D. 2013 Wing venation homologies in Pennsylvanian ‘cockroachoids' (Insecta) clarified thanks to a remarkable specimen from the Pennsylvanian of Ningxia (China). J. Syst. Palaeontol. 11, 41-46. (doi:10.1080/14772019.2011.637519).

Kukalova-Peck, J. & Lawrence, J.F. 2004 Relationships among coleopteran suborders and major endoneopteran lineages: Evidence from hind wing characters. Eur. J. Entomol. 101, 195-144.

Li, X.-R., Zheng, Y.-H., Wang, C.-C. & Wang, Z.-Q. 2018 Old method not old-fashioned: parallelism between wing venation and wing-pad tracheation of cockroaches and a revision of terminology. Zoomorphology. (doi:10.1007/s00435-018-0419-6).

Brannoch, S.K., Wieland, F., Rivera, J., Klass, K.-D., Béthoux, O. & Svenson, G.J. 2017 Manual of praying mantis morphology, nomenclature, and practices (Insecta, Mantodea). ZooKeys 696, 1-100. (doi:10.3897/zookeys.696.12542).

Validity of the findings

The data appears robust, although the illustrations could be presented in higher resolution (although perhaps this is due to the PDF generation process...I did not review individual files).

The conclusions are somewhat, but not entirely, justified given the data. In the taxonomic section, justification for the establishment of the new genus is not given, and differentiation with Qilianiblatta and Qilianiblatta namurensis is not defined. The authors also do not acknowledge literature (Ross 2012) discussing reliability and conservation of venation character states in tegminized forewings, which could be highly relevant to how meaningful their results (and the results of other taxonomic works, like that describing Qilianiblatta) are. Also, i think the revision of the age of the geoloigical setting based on this newly described fossil are not strongly justified and actually detract from the quality of the paper (see further explanation in attached doc).

---

## Round 0.2 · Minor Revisions

Dear Drs. Calisto and Piñeiro:

Thanks for re-submitting your manuscript to PeerJ, and for addressing the concerns raised by the reviewers. I have now received one independent review of your revision, and as you will see, it is mostly favorable. Well done! Nonetheless, the reviewer raised some relatively minor concerns about the research, and areas where the manuscript can still be improved. I agree with the reviewer, and thus feel that these concerns should be adequately addressed before moving forward.

Therefore, I am recommending that you revise your manuscript accordingly, taking into account all of the issues raised by the reviewers. I do believe that your manuscript will be ready for publication once these issues are addressed.

Good luck with your revision,

-joe

·

Basic reporting

'no comment'

Experimental design

'no comment'

Validity of the findings

'no comment'

Additional comments

The paper is now after revision acceptable (after regarding the second review in attached pdf). General comments are to compare the shape of the wing (as it differs from Quilianblatta) with Neorthroblattina and possibly others. Still lacking is the calculation of forewing area, which is easily possible. Meanwhile few papers on fossil cockroaches (notice at least those published in 2017-2019 on amber to see the general knowledge of this group) were published and this as honor to other students dealing with cockroaches should be included at least in introduction. This shifts the whole field. Fortunately I see the references by Schneider were considered after first revision.

---

## Round 0.3 · accepted · Accept

Dear Drs. Calisto and Piñeiro:

Thanks for re-submitting your manuscript to PeerJ, and for addressing the concerns raised by the reviewers. I now believe that your manuscript is suitable for publication. Congratulations! I look forward to seeing this work in print, and I anticipate it being an important resource for entomologists, particularly those studying cockroach systematics and evolution. Thanks again for choosing PeerJ to publish such important work.

-joe